# PeFoo-L: A General Framework for Preconditioned Forward-Only Optimizer Enabling LLM Fine-tuning on the Edge

## Abstract

Fine-tuning Large Language Models (LLMs) on resource-constrained edge devices is a critical but challenging task, primarily due to the prohibitive memory and computational costs of backpropagation. While forward-only optimizers like MeZO mitigate these costs by eliminating the backward pass, they often suffer from slow and unstable convergence, particularly on loss landscapes with heterogeneous curvature. To address this limitation, we introduce PeFoo, a general framework for preconditioner enhanced forward only optimizer. PeFoo integrates a carefully designed preconditioning strategy into the forward-only paradigm, corrects a fundamental source of bias and instability present in prior work HiZOO. Furthermore, to counteract the memory overhead introduced by the preconditioner itself, we propose PeFoo-L, which employs a layer-wise update strategy. This approach constrains preconditioner storage and weight updates to a single layer per iteration, reducing the overall memory footprint and data traffic. Experimental results validate the effectiveness of our framework. On the OPT-1.3B model, PeFoo surpasses the accuracy of leading zeroth-order methods MeZO and HiZOO by 2.7% and 2.1%, respectively. Furthermore, PeFoo-L achieves a memory footprint reduction of over $2.73\times$ and $1.75\times$ compared to Adam and HiZOO, while delivering faster convergence speed compared to MeZO and HiZOO.

## 1 Introduction

In recent years, the increasing number of applications for Transformers requires the training and fine-tuning of these Transformer models on edge platforms (Lee & Yoo, 2021). Real-world scenarios often demand that models be retrained or fine-tuned using personal or domain-specific data. The transmission of sensitive data to centralized servers for this purpose introduces latency and privacy risks, thereby emphasizing the critical need for on-device training solutions.

However, conventional Transformer training is dominated by the backpropagation algorithm, a process requiring both forward and backward passes through the neural network. This dual-pass mechanism is resource-intensive, requiring substantial memory overhead from gradient storage and demanding extensive computational power for derivative calculations, which makes it prohibitive for deployment on edge devices. This software-level challenge is mirrored in hardware design; the majority of existing Application-Specific Integrated Circuit (ASIC) or Field-Programmable Gate Array (FPGA) Transformer accelerators (*e.g.*, ELSA (Ham et al., 2021), DOTA (Qu et al., 2022), FACT (Qin et al., 2023)) are optimized exclusively for inference. They typically omit backpropagation capabilities due to the design complexities of co-locating them with forward-pass operations, a decision that severely limits the utility of these accelerators for on-device learning.

In response to these challenges, recent work has pioneered forward-only optimization strategies, most notably MeZO (Malladi et al., 2023). By avoiding the backward pass entirely, MeZO substantially reduces memory consumption, making it feasible to fine-tune large scale models on memory constrained devices. However, this memory efficiency comes at a cost of slow and unstable convergence. To accelerate convergence, HiZOO (Zhao et al., 2025) makes a significant stride by incorporating estimated Hessian information into the MeZO framework. However, our analysis reveals that HiZOO's Hessian estimator is inherently biased due to the premature application of an abso-

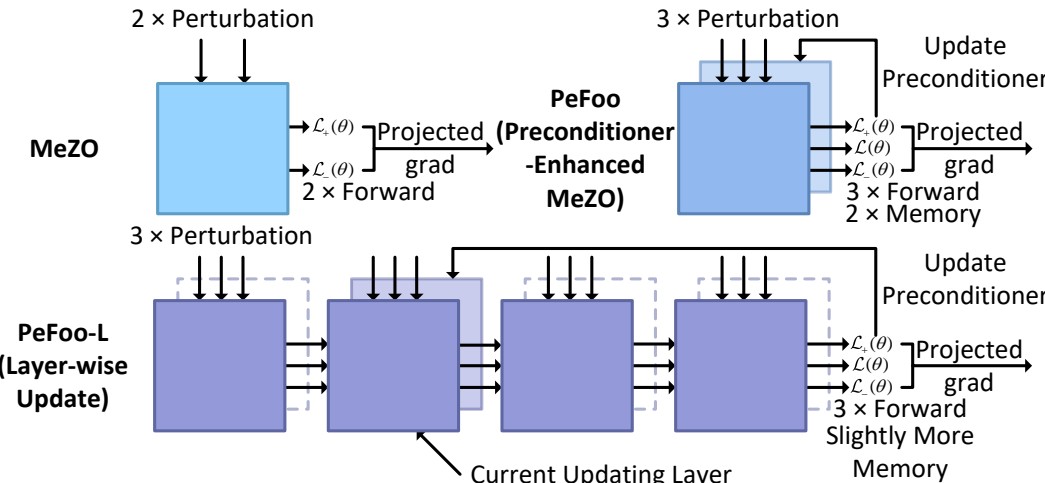

Figure 1: Schematic of the MeZO and proposed PeFoo, PeFoo-L training pipeline. Updating the preconditioner demands an additional forward pass and requires storage capacity equivalent to the model parameters. In contrast, the layer-wise update method employed by PeFoo optimizes memory efficiency by retaining preconditioners for only one block during the fine-tuning process.

lute value function, and its weight update rule is inconsistent with its gradient formulation. These fundamental issues lead to numerical instability, particularly under FP16 precision, and can trap the optimizer in suboptimal convergence paths.

Furthermore, incorporating conventional Hessian information into a forward-only framework presents a fundamental dilemma: their associated storage overhead directly counteracts the memory efficiency gains that make methods like MeZO attractive in the first place. This inherent trade off between convergence speed and memory footprint forms a significant barrier to deploying high performance, preconditioned optimizers on resource constrained edge devices.

To address these critical shortcomings and unlock the true potential of second-order information in ZO methods, we introduce **PeFoo**, a general framework for preconditioner enhanced forward only optimizer built upon a principled correction of HiZOO. To specifically counter the memory overhead introduced by the preconditioner itself, we further propose **PeFoo-L**, a variant that incorporates a hardware-friendly, layer-wise weight update strategy. This mechanism computes preconditioners for a single layer per iteration, thereby preserving high memory efficiency and rapid convergence.

The primary contributions of this paper are as follows:

**Theoretically-Grounded Preconditioning for Edge Fine-tuning.** We introduce **PeFoo**, a preconditioned forward only optimization framework designed for robustness and efficiency. Its innovation is an unbiased Hessian estimator, which corrects a fundamental source of bias and instability present in prior work like HiZOO. By preserving negative curvature information, PeFoo ensures both stable and rapid convergence.

**Hardware-Friendly Edge Fine-tuning Framework.** We analyze the inefficiency in preconditioner enhanced zeroth order optimizer and thus propose **PeFoo-L**, a layer-wise update strategy that optimizes the preconditioner's memory footprint. By localizing updates, this method minimizes storage requirements and reduces memory traffic between on-chip cache and DRAM, making it particularly suitable for deployment on edge hardware accelerators.

**Comprehensive Evaluation.** We conduct extensive experiments across multiple benchmark datasets to validate the efficacy and generalizability of our methods. The results demonstrate that PeFoo outperforms zeroth-order methods MeZO and HiZOO by 2.7% and 2.1% in average accuracy on OPT-1.3B, respectively. Furthermore, PeFoo-L achieves a memory footprint reduction of over $2.73\times$ and $1.75\times$ compared to Adam and HiZOO, while delivering faster convergence speed than MeZO and HiZOO.

## 2 BACKGROUND

### 2.1 PRELIMINARIES OF ZEROTH-ORDER OPTIMIZERS

#### 2.1.1 MEZO

Recently, MeZO (Malladi et al., 2023) firstly adapted the classical ZO-SGD method to fine-tune LLMs, achieving comparable performance with significant memory reduction. Consider a labelled dataset $\mathcal{D} = \{(x_i, y_i)\}_{i \in [|\mathcal{D}|]}$ and a minibatch $\mathcal{B} \subset \mathcal{D}$ of size $B$, we let $\mathcal{L}(\theta; \mathcal{B})$ denote the loss on the minibatch. Given a model with parameters $\theta \in \mathbb{R}^d$ and a loss function $\mathcal{L}$, SPSA (Spall, 1992) estimates the gradient on a minibatch $\mathcal{B}$ as

$$\widehat{\nabla}\mathcal{L}(\theta; \mathcal{B}) = \frac{\mathcal{L}(\theta + \lambda p; \mathcal{B}) - \mathcal{L}(\theta - \lambda p; \mathcal{B})}{2\lambda} p \approx pp^{\top}\nabla\mathcal{L}(\theta; \mathcal{B}), \tag{1}$$

where $p \in \mathbb{R}^d$ with $p \sim \mathcal{N}(0, I_d)$ and $\lambda$ is the perturbation scale. The n-SPSA gradient estimate averages $\widehat{\nabla}\mathcal{L}(\theta; \mathcal{B})$ over $n$ randomly sampled $p$.

ZO-SGD is an optimizer with learning rate $\eta$ that updates parameters as $\theta_t = \theta_{t-1} - \eta\widehat{\nabla}\mathcal{L}(\theta; \mathcal{B}_{t-1})$, where $\mathcal{B}_t$ is the minibatch at time $t$ and $\widetilde{\nabla}\mathcal{L}$ is the SPSA gradient estimate.

### 2.2 RELATED WORK

#### 2.2.1 ZEROTH-ORDER OPTIMIZATION FOR LLM

MeZO (Malladi et al., 2023) is the first to demonstrate that forward-only updates can fine-tune LLMs to high accuracy while reducing GPU memory. MeZO's success has inspired further research, including variants that incorporate sparsity (Guo et al., 2024; Liu et al., 2024) and the creation of extensive benchmarks for zeroth-order (ZO) fine-tuning methods (Zhang et al., 2024). ReLIZO (Wang et al., 2024) is a zeroth-order optimizer that reuses historical query samples by modeling gradient estimation as a quadratically constrained linear program, which reduces computation complexity while maintaining efficacy. To accelerate convergence, HiZOO (Zhao et al., 2025) utilizes estimated Hessian information within the ZO framework. However, this enhancement comes at the cost of doubling the required memory, which counteracts the primary benefit of forward-only methods. B-PDF (Yu et al., 2024) combines the block coordinate descent (BCD) method with a Hessian-informed zeroth-order optimizer, yet provides limited accuracy improvement and acceleration in convergence compared to MeZO. In contrast to previous methods, our method prioritizes accuracy improvement by benchmarking and enhancing preconditioners, while remaining optimized for the resource constraints of edge computing.

#### 2.2.2 PRECONDITIONERS FOR STOCHASTIC TRAINING

High curvature directions in SGD require a small learning rate to avoid overshooting, leading to slow progress in low curvature directions. Preconditioning (Dauphin et al., 2015; Qu et al., 2024) enhances optimization convergence and efficiency by regulating curvature, particularly in SGD. Shampoo (Gupta et al., 2018) maintains a preconditioning matrix for each dimension of a weight tensor, which is updated using the second-moment statistics of accumulated gradients. AdaBelief (Zhuang et al., 2020) introduces an adaptive preconditioner that scales the stepsize according to the observed gradient. AdaHessian (Yao et al., 2021) and Sophia (Liu et al., 2023) are adaptive second-order optimizers that estimate the diagonal Hessian as its preconditioner. However, preconditioner introduces additional storage requirements and increases memory access times, which makes it less practical in resource-constrained scenarios.

## 3 METHODOLOGY

### 3.1 PRECONDITIONER ENHANCED OPTIMIZER: PEFOO

Our approach begins by applying a preconditioner matrix $D$ through a linear transformation of the parameter space, $\widetilde{\theta} = D^{\frac{1}{2}}\theta$. The resulting descent direction for the zeroth-order gradient estimate is

---

**Algorithm 1** Training pipeline of proposed PeFoo.

---

**Require:** parameters $\theta \in \mathbb{R}^d$, loss function $\mathcal{L}$, perturbation scale $\lambda$, learning rate $\eta$, smooth scale $\alpha$

1: $D_0 \leftarrow I$ ▷ Preconditioner Initialization
2: **for** $t = 1, \ldots, T$ **do**
3:     Sample a random seed $s$
4:     $\ell \leftarrow \mathcal{L}(\theta; \mathcal{B})$
5:     $\theta \leftarrow \text{PerturbParameters}(\theta, \lambda, D_{t-1}^{-1/2}, s)$
6:     $\ell_\lambda \leftarrow \mathcal{L}(\theta; \mathcal{B})$
7:     $\theta \leftarrow \text{PerturbParameters}(\theta, -2\lambda, D_{t-1}^{-1/2}, s)$
8:     $\ell_{-\lambda} \leftarrow \mathcal{L}(\theta; \mathcal{B})$
9:     $\theta \leftarrow \text{PerturbParameters}(\theta, \lambda, D_{t-1}^{-1/2}, s)$ ▷ Reset Parameters Before Descent
10:     projected_grad $\leftarrow \frac{1}{2\lambda}(\ell_\lambda - \ell_{-\lambda})D_{t-1}^{-1/2}$
11:     Reset random number generator with seed $s$
12:     **for** $\theta_i \in \theta$ **do**
13:         Sample $p_i \sim \mathcal{N}_s(0, I)$
14:         $\theta_i \leftarrow \theta_i - \eta_t \times \text{projected\_grad}_i \times p_i$ ▷ Update Weights
15:     **end for**
16:     $\widehat{H}_t \leftarrow \frac{1}{2\lambda^2}(\ell_\lambda + \ell_{-\lambda} - 2\ell)\left(D_{t-1}^{1/2} p_i p_i^\top D_{t-1}^{1/2} - D_{t-1}\right)$ ▷ **Unbiased Hessian Estimator**
17:     $D_t \leftarrow f(\widehat{H}_t)$ ▷ **Post-Weight Preconditioner Update**
18: **end for**

19: **function** PERTURBPARAMETER$(\theta, \lambda, D_t^{-1/2}, s)$
20:     Reset random number generator with seed $s$
21:     **for** $\theta_i \in \theta$ **do**
22:         Sample $p_i \sim \mathcal{N}_s(0, I)$
23:         $\theta_i \leftarrow \theta_i + \lambda D_t^{-1/2} p_i$ ▷ Perturb Parameters With Preconditioner
24:     **end for**
25:     **return** $\theta$
26: **end function**

---

derived as:

$$\widehat{\nabla}\mathcal{L}(\theta; \mathcal{B}) = D^{-1/2}\widehat{\nabla}\widetilde{\mathcal{L}}(\widetilde{\theta}_t; \mathcal{B})$$
$$= \frac{\mathcal{L}\left(\theta + \lambda D^{-1/2}p; \mathcal{B}\right) - \mathcal{L}\left(\theta - \lambda D^{-1/2}p; \mathcal{B}\right)}{2\lambda} \cdot D^{-1/2}p. \tag{2}$$

The key insight from Eq. 2 is that the perturbation $\lambda D^{-1/2}p$ is no longer isotropic. This adaptive scaling ensures that the exploration of the loss surface is aligned with its local geometry, enabling more efficient optimization.

A critical challenge, however, is the selection of an appropriate preconditioner $D$. Drawing inspiration from quasi-Newton methods, we leverage an unbiased Hessian estimator $\widehat{H}$ from HiZOO (Zhao et al., 2025):

$$\widehat{H} = \nabla^2\mathcal{L}(\theta) = \frac{1}{2}\left[\frac{\Delta\mathcal{L}}{\lambda^2} \cdot \left(D^{1/2}pp^\top D^{1/2} - D\right)\right], \tag{3}$$

where $\Delta\mathcal{L} = \mathcal{L}(\theta + \lambda D^{-1/2}p; \mathcal{B}) + \mathcal{L}(\theta - \lambda D^{-1/2}p; \mathcal{B}) - 2\mathcal{L}(\theta; \mathcal{B})$. Note that this derivation permits the use of any symmetric positive semi-definite preconditioner $D$ without structural constraints. We prove this in Appendix B. This flexibility permits a broad class of preconditioners of the form $D = f(\widehat{H})$, significantly expanding the design space.

For PeFoo, we introduce a specific preconditioner: $f(\widehat{H}_t) = \text{clip}(\text{abs}(\text{EMA}(\widehat{H})), D_{min}, D_{max})$. We set $D_{min}=1e\text{-}1$ and $D_{max}=1e4$ in our practical use. We restrict our implementation to diagonal matrices, which reduces memory complexity from $\mathcal{O}(d^2)$ to $\mathcal{O}(d)$ while maintaining strong empiri-

cal performance and ensuring the preconditioner is positive semi-definite. Our design decouples the estimation process from the preconditioner construction.

Two key factors can summarize the main difference between PeFoo and HiZOO: 1) **Unbiased estimation of the Hessian matrix**: PeFoo uses $\hat{H}_t = \frac{1}{2\lambda^2}\left(\ell_\lambda + \ell_{-\lambda} - 2\ell\right)\left(D_{t-1}^{1/2} p_i p_i^\top D_{t-1}^{1/2} - D_{t-1}\right)$ to perform the EMA update. In contrast, HiZOO uses $H_t^{-1} = \frac{1}{2\lambda^2}|\ell_\lambda + \ell_{-\lambda} - 2\ell| H_{t-1}^{-1/2} p_i p_i^\top H_{t-1}^{-1/2}$. Consequently, HiZOO's estimation tends to be larger than PeFoo's unbiased estimation and does not account for the possibility of negative elements in the Hessian matrix, leading to saddle point attraction artifacts. 2) **Weight update step**: HiZOO uses the already updated Hessian matrix $H_t$ in its weight update step, rather than $H_{t-1}$. This approach is inconsistent with the gradient calculation in Eq. 2, which uses $D_{t-1}$. This inconsistency is the reason for the large variance in HiZOO's Hessian matrix estimation, which often results in extreme values that overflow the boundaries of FP16.

The full training pipeline for PeFoo is detailed in Algorithm 1. At each iteration, the algorithm performs three forward passes to evaluate the loss function $\mathcal{L}(\theta)$: once at the current parameters, and twice more after perturbing them. The resulting loss values are used to compute a projected gradient estimate using the preconditioner from the previous step ($D_{t-1}$). After the model weights are updated, the same loss values are used to form an estimate of the Hessian $\hat{H}_t$, which in turn updates the preconditioner for the next iteration via $f(\hat{H}_t) = \text{clip}(\text{abs}(\text{EMA}(\hat{H})), D_{min}, D_{max})$.

## 3.2 CONVERGENCE ANALYSIS

We analyze the convergence of PeFoo under the following standard assumptions:

1. The objective function $\mathcal{L}(\theta)$ is $L$-smooth, i.e. $\mathcal{L}(\theta_{t+1}) \leqslant \mathcal{L}(\theta_t) - \langle \nabla\mathcal{L}(\theta_t), \theta_{t+1} - \theta_t \rangle + \frac{L}{2}\|\theta_{t+1} - \theta_t\|^2$;

2. The stochastic gradient $\nabla\mathcal{L}(\theta; \mathcal{B})$ has $\sigma^2$ variance, i.e. $\mathbb{E}\left[\|\nabla\mathcal{L}(\theta; \mathcal{B}) - \nabla\mathcal{L}(\theta)\|^2\right] \leqslant \sigma^2$;

3. $D = \text{diag}(d_1, \ldots, d_n)$, with $0 < \beta_\ell \leqslant d_i \leqslant \beta_u$.

After $T$ iterations with the update rule $\theta_t = \theta_{t-1} - \eta\widehat{\nabla}\mathcal{L}(\theta; \mathcal{B}_{t-1})$, the average expected squared norm of the gradient is bounded as follows:

$$\mathbb{E}\left[\frac{1}{T}\sum_{t=1}^{T}\|\nabla\mathcal{L}(\theta_t; \mathcal{B})\|^2\right] \leqslant \frac{\beta_u}{T}\sum_{t=1}^{T}\|\nabla\mathcal{L}(\theta_t; \mathcal{B})\|_{D_t^{-1}}^2$$

$$\leqslant \frac{2\beta_u(\mathcal{L}(\theta_0) - \mathcal{L}^*)}{\eta T} + L\beta_u\eta\sigma^2 + \mathcal{O}(\lambda^2).$$

Let $\eta \to 0$ as $T \to \infty$ (e.g., $\eta = \mathcal{O}(1/\sqrt{T})$), we conclude that:

$$\lim_{T \to \infty}\mathbb{E}\left[\|\nabla\mathcal{L}(\theta; \mathcal{B})\|^2\right] = 0.$$

*Proof.* Detailed proof can be found in Appendix C. □

It is important to highlight the implications of Assumption 3. The requirement for a bounded preconditioner ($d_i \leqslant \beta_u$) is necessary to establish the convergence rate's upper bound. This theoretical condition underscores the practical importance of constraining the values of the preconditioner during training to ensure stability.

## 3.3 THE BOTTLENECKS OF PRECONDITIONED ZEROTH-ORDER METHODS

While preconditioners offer significant convergence acceleration, they introduce a substantial memory cost that scales with the number of model parameters $d$: $\mathcal{O}(d)$ for a diagonal matrix and $\mathcal{O}(d^2)$ for a full one. As detailed in Table 1, this additional overhead can be severe, particularly for large models on edge devices. This storage requirement directly undermines the primary objective of forward-only methods like MeZO to achieve memory-efficient optimization.

Table 1: Peak GPU memory usage (GB) for fine-tuning OPT models on SST-2 (FP16 precision). OOM: out of memory.

| Device | Model(Parameters) | Adam | MeZO | HiZOO | PeFoo | PeFoo-L |
|---|---|---|---|---|---|---|
| RTX 4090 (24GB) | OPT-1.3B (2.45GB) | 10.22GB | 3.17GB | 6.06GB | 6.27GB | **3.46GB** |
| | OPT-6.7B (12.41GB) | OOM | 14.33GB | OOM | OOM | **14.76GB** |
| Tesla A100 (80GB) | OPT-6.7B (12.41GB) | 50.39GB | 14.33GB | 27.11GB | 27.50GB | **14.76GB** |
| | OPT-13B (23.94GB) | OOM | 26.35GB | 50.77GB | 51.37GB | **26.63GB** |

Table 2: Average memory traffic on RTX 4090 per training step (GB) on OPT-1.3B. Lower values are better.

| | Forward Pass Traffic | | Weight Update Traffic | |
|---|---|---|---|---|
| Method | L2 ↔ L1 | L2 ↔ DRAM | L2 ↔ L1 | L2 ↔ DRAM |
| MeZO | 57.41 | 9.58 | 100.49 | 58.53 |
| PeFoo | 86.12 | 14.38 | 249.99 | 144.48 |
| PeFoo-L | 86.12 | 14.38 | **28.96** | **17.75** |

Furthermore, beyond static memory costs, the parameter update step itself emerges as a significant performance bottleneck. Unlike the forward pass, which benefits from high data reuse and effective use of on-chip caches, the weight update process is characterized by poor cache locality. The weight update process requires frequent, high-volume data transfers between the power-hungry system DRAM and the limited on-chip L2 cache, a phenomenon quantified in Table 2. This issue is intensified on edge devices where DRAM bandwidth is an insufficient resource (*e.g.*, a DDR4 module offers a peak bandwidth of only 25.6 GB/s, far below the TB/s-level throughput of on-chip SRAM). Alleviating this memory traffic bottleneck is therefore critical for enabling efficient on-device training.

### 3.4 LAYER-WISE WEIGHT UPDATE: PEFOO-L

To address the dual challenges of memory traffic and preconditioner storage, we introduce PeFoo-L, a hardware-friendly, layer-wise weight update strategy. This approach, which shares conceptual similarities with other layer-wise methods like BAdam (Luo et al., 2024), LiSA (Pan et al., 2024), and B-PDF (Yu et al., 2024), is integrated directly into our preconditioned forward-only framework. As illustrated in Fig. 1, PeFoo-L selectively updates only a subset of model layers during each optimization step, leaving the remaining layers static. This targeted update strategy yields a crucial benefit: by confining preconditioner computations exclusively to the active layer, it dramatically reduces the memory required for their storage.

We present the pseudo-code for the proposed optimizer combined with the layer-wise weight update algorithm in Appendix D. To optimize memory utilization, in each iterative step, we update the weights of only one layer, and the preconditioner estimate is reinitialized upon completing updates. The sequence of the selected layer is from the last layer to the first layer, which is empirically better than from the first layer to the last layer. Detailed proof of PeFoo-L's convergence is provided in Appendix D.

## 4 EXPERIMENTS

### 4.1 EXPERIMENTAL SETTINGS

**Dataset & Baselines.** To evaluate both the accuracy and training efficiency of PeFoo, we adopt the experimental setup of MeZO. Our experiments are conducted using RoBERTa-large (Liu, 2019) and OPT-1.3B (Zhang et al., 2023) on widely used datasets from the GLUE and SuperGLUE benchmark (Wang et al., 2018; 2019). For both training and validation on RoBERTa-large, we set $k$=16, meaning we have 16 examples per class. For experiments involving the OPT-1.3B model, we randomly

Table 3: Performance comparison on OPT-1.3B (with 1000 examples) using MeZO, HiZOO, PeFoo, PeFoo-L and Adam (FT). ICL: in-context learning; LP: linear probing. PEFT represents the LoRA and prefix and we report the best of them. All FT experiments train for 5 epochs, and MeZO, HiZOO, PeFoo, PeFoo-L use 20K steps.

| Task | SST-2 | RTE | CB | BoolQ | WSC | WIC | COPA | ReCoRD | SQuAD | Average |
|------|-------|-----|-----|-------|-----|-----|------|--------|-------|---------|
| Task type | | | classification | | | | – multiple choice – | | generation | |
| Zero-shot | 53.6 | 53.4 | 37.5 | 45.5 | 43.3 | 57.1 | 75.0 | 70.6 | 27.1 | 51.5 |
| ICL | 80.3 | 53.1 | 48.2 | 58.5 | 45.2 | 50.9 | 69.0 | 71.1 | 58.9 | 59.5 |
| LP | 91.1 | 61.4 | 64.3 | 62.4 | 63.5 | 62.1 | 43.0 | 18.1 | 3.2 | 52.1 |
| MeZO | 91.7 | 64.3 | 69.6 | 65.5 | **63.5** | 57.7 | 77.0 | 71.2 | 75.8 | 70.7 |
| MeZO(PEFT) | 91.2 | 67.1 | 71.4 | 63.8 | 60.6 | 58.0 | **77.0** | 71.7 | 75.7 | 70.7 |
| HiZOO | 91.7 | 64.3 | 71.4 | 65.5 | **63.5** | 57.7 | 78.0 | 71.4 | 78.2 | 71.3 |
| HiZOO(PEFT) | 89.2 | 67.5 | 71.4 | 64.7 | **63.5** | 60.6 | 75.0 | 71.1 | 76.1 | 71.0 |
| PeFoo | 91.7 | **66.1** | **83.9** | 65.8 | **63.5** | 59.4 | **80.0** | 71.5 | **78.4** | **73.4** |
| PeFoo(PEFT) | **91.3** | **68.2** | **73.2** | 65.6 | 59.6 | 58.8 | 77.0 | **72.1** | **77.5** | **71.4** |
| PeFoo-L | **91.9** | 65.3 | 73.2 | **66.2** | **63.5** | 59.3 | 76.0 | **72.0** | 77.5 | 71.6 |
| PeFoo-L(PEFT) | 90.6 | 63.5 | 71.4 | 64.0 | 59.6 | 59.7 | 72.0 | 71.2 | 74.5 | 69.6 |
| FT | 94.4 | 75.8 | 89.3 | 74.9 | 62.5 | 65.2 | 79.0 | 71.9 | 83.6 | 77.4 |

Table 4: Performance comparison on RoBERTa-large (350M parameters, sample numbers $k = 16$) using MeZO, variants of PeFoo, and Adam (FT). PEFT represents the LoRA and prefix and we report the best of them. All FT experiments employs 1K steps, MeZO employs 100K steps, and HiZOO, PeFoo and PeFoo-L use 5K steps. All reported numbers are averaged accuracy (standard deviation) across 5 runs.

| Task | SST-2 | SST-5 | SNLI | MNLI | RTE | TREC | Average |
|------|-------|-------|------|------|-----|------|---------|
| Task type | —— sentiment —— | | —— natural language inference —— | | | – topic – | |
| Zero-shot | 79.0 | 35.5 | 50.2 | 48.8 | 51.4 | 32.0 | 49.5 |
| LP | 76.0 ($\pm$2.8) | 40.3 ($\pm$1.9) | 66.0 ($\pm$2.7) | 56.5 ($\pm$2.5) | 59.4 ($\pm$5.3) | 51.3 ($\pm$5.5) | 58.3 |
| MeZO | 90.5 ($\pm$1.2) | 45.5 ($\pm$2.0) | 68.5 ($\pm$3.9) | 58.7 ($\pm$2.5) | 64.0 ($\pm$3.3) | **76.9** ($\pm$2.7) | 67.4 |
| MeZO(PEFT) | 91.4 ($\pm$0.9) | 45.8 ($\pm$2.0) | 71.6 ($\pm$2.5) | 64.0 ($\pm$2.5) | 65.4 ($\pm$3.9) | **80.3** ($\pm$3.6) | 69.8 |
| HiZOO | 91.5 ($\pm$1.2) | 44.3 ($\pm$1.2) | **70.7** ($\pm$3.1) | 62.6 ($\pm$0.9) | 65.2 ($\pm$2.1) | 73.6 ($\pm$4.6) | 68.0 |
| HiZOO(PEFT) | 91.6 ($\pm$0.8) | 45.1 ($\pm$2.2) | 66.1 ($\pm$3.0) | 62.5 ($\pm$1.3) | **66.2** ($\pm$2.5) | 70.1 ($\pm$5.4) | 66.9 |
| PeFoo | **91.7** ($\pm$1.4) | **46.1** ($\pm$0.8) | 70.3 ($\pm$2.9) | **64.7** ($\pm$1.6) | 65.6 ($\pm$2.1) | 74.0 ($\pm$4.3) | **68.7** |
| PeFoo(PEFT) | **92.3** ($\pm$1.1) | **46.0** ($\pm$1.2) | 72.0 ($\pm$1.6) | **64.9** ($\pm$1.2) | 66.0 ($\pm$2.6) | 79.7 ($\pm$3.3) | **70.2** |
| PeFoo-L | 91.5 ($\pm$0.8) | 45.1 ($\pm$1.7) | 70.3 ($\pm$1.9) | 62.6 ($\pm$1.1) | **66.2** ($\pm$1.6) | 73.8 ($\pm$4.6) | 68.2 |
| PeFoo-L(PEFT) | 91.8 ($\pm$0.7) | 45.5 ($\pm$2.0) | **72.2** ($\pm$2.6) | 63.7 ($\pm$1.2) | 65.2 ($\pm$3.1) | 74.3 ($\pm$4.8) | 68.8 |
| FT | 91.9 ($\pm$1.8) | 47.5 ($\pm$1.9) | 77.5 ($\pm$2.6) | 70.0 ($\pm$2.3) | 66.4 ($\pm$7.2) | 85.0 ($\pm$2.5) | 73.1 |

sample 1,000 examples for training, 500 examples for validation, and 1,000 examples for testing. All models are trained using FP16 precision to simulate the constraints of edge device scenarios. All accuracy evaluations are performed on a single NVIDIA RTX 4090 (24GB) GPU. Detailed hyperparameter configurations used in our experiments are summarized in Appendix A.

For our comparative experiments, we noted that the official HiZOO implementation encounters numerical instability when operating under FP16 precision, frequently leading to overflow issues that halt the training process. To enable a stable and meaningful comparison, we introduced a minimal necessary modification by clamping its Hessian estimator to a maximum value of $1e4$. This change specifically addresses the overflow problem while preserving the core algorithmic logic presented in the original HiZOO paper.

## 4.2 ACCURACY AND CONVERGENCE SPEED

Tables 3 and 4 present the experimental results for our proposed frameworks, PeFoo and PeFoo-L, in comparison with the MeZO baseline, HiZOO, and traditional fine-tuning using the Adam optimizer (FT) on the OPT-1.3B model and the RoBERTa-large model. We highlight the best results for full-parameter tuning and PEFT, respectively. As shown in Tables 3 and 4, PeFoo and PeFoo-L across full-parameter tuning, LoRA and prefix, outperform Zero-shot, ICL, and LP on all datasets

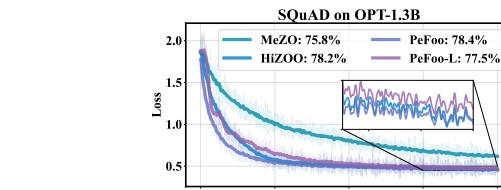
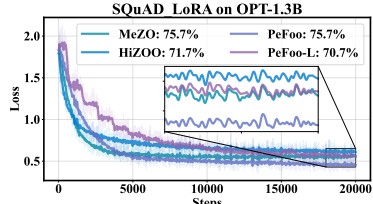

Figure 2: Steps-based loss curves of using MeZO, HiZOO, and proposed PeFoo and PeFoo-L.

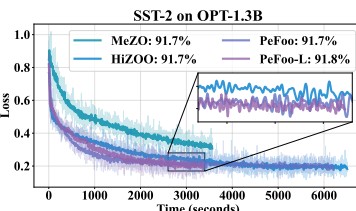
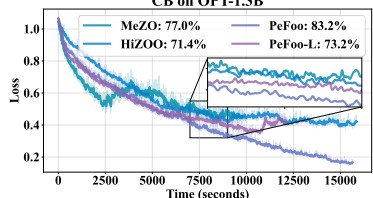

Figure 3: Time-based loss curves of using MeZO, HiZOO, PeFoo and PeFoo-L on RTX 4090.

except WIC. PeFoo also outperforms MeZO by an average of 2.7%, HiZOO by 2.1% in OPT-1.3B absolute accuracy, and outperforms MeZO by 1.3% and HiZOO by 0.7% in RoBERTa-large absolute accuracy. Figs. 2 and 3 show the step-based and time-based loss curves, respectively, which indicate that PeFoo demonstrates substantially faster convergence in terms of training steps compared to MeZO and HiZOO within 20,000 steps for OPT-1.3B.

## 4.3 EXECUTION SPEED ANALYSIS

As shown in Fig. 3, PeFoo and HiZOO require approximately $2\times$ the wall-clock time of MeZO and PeFoo-L. Contrary to HiZOO's claim that this is mainly due to an extra forward pass, our profiling with NVIDIA Nsight Compute identifies the weight update stage as the true performance bottleneck, which accounts for 71% of the total cycle count. While the forward pass contributes a $1.5\times$ overhead, our layer-wise method, PeFoo-L, accelerates the weight update by $3.26\times$, effectively closing the performance gap with MeZO. This demonstrates that optimizing the weight update is more critical for execution speed than minimizing forward passes in this class of optimizers.

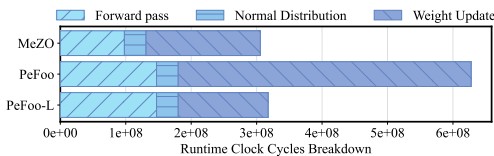
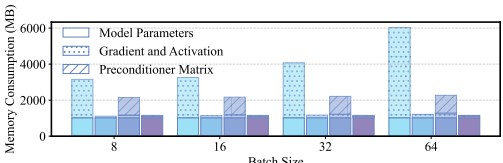

Figure 4: Training runtime clock cycles breakdown for MeZO, PeFoo, and PeFoo-L frameworks for training OPT-1.3B on SST-2 dataset. The analysis reveals that weight updates constitute bottlenecks in the PeFoo's training process.

Figure 5: Memory consumption of different components for fine-tuning the RoBERTa-large model on SST-2 dataset using different methods. The left to right is Adam, MeZO, HiZOO or PeFoo, PeFoo-L, respectively.

## 4.4 MEMORY CONSUMPTION ANALYSIS

Fig. 5 illustrates a comparative analysis of memory consumption across several optimizers. Our empirical assessment of GPU memory utilization revealed that PeFoo-L exhibits a memory cost comparable to MeZO and reduces memory footprint by $1.84\times$ compared to HiZOO or PeFoo and more than $2.73\times$ compared to Adam. In comparison, HiZOO demonstrates a 84% higher memory

cost than MeZO, indicating an impractical convergence-memory trade-off in scenarios demanding low memory usage.

### 4.5 ABLATION STUDY OF PRECONDITIONER

We conduct ablation studies on the SST-2 dataset to analyze and validate the key design choices for our preconditioner.

#### 4.5.1 IMPACT OF PRECONDITIONER CLAMPING

We investigate the effect of constraining the preconditioner's values, a practice suggested by our convergence analysis. As shown in Fig. 6, applying a clamp results in more stable convergence compared to using an unbounded preconditioner. Specifically, clamping the maximum value to 1.0 or 0.1 improves absolute accuracy by 0.8% and 1.1%, respectively, confirming the practical benefits of this technique.

#### 4.5.2 EFFECTIVENESS OF DIFFERENT PRECONDITIONERS

We validate the effectiveness of our specific preconditioner formulation, $D = \mathrm{abs}(\mathrm{EMA}(\hat{H}))$. We compare it against two primary baselines: an identity preconditioner, $D = I$ (which is equivalent to MeZO), and an alternative formulation, $D = \mathrm{EMA}(\mathrm{abs}(\hat{H}))$, which applies the absolute value before the moving average. The results in Fig. 6 demonstrate the clear superiority of our proposed design. PeFoo exhibits significantly faster convergence speed than both baselines and achieves an absolute accuracy gain of 2.6% over the $D = \mathrm{EMA}(\mathrm{abs}(\hat{H}))$ formulation. This validates our choice to maintain sign-aware Hessian during the EMA process.

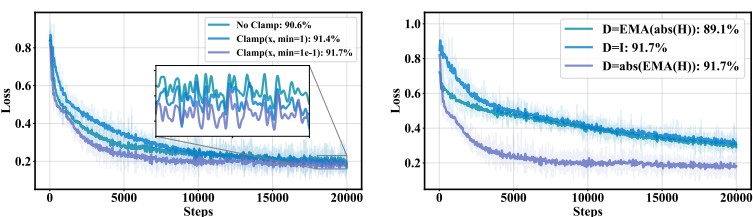

Figure 6: Ablation Study of Preconditioners on the SST-2 Dataset. (Left): The effect of clamping the preconditioner on the training loss. (Right): A comparison of training loss curves for different preconditioner choices.

## 5 CONCLUSION

Our work demonstrates that PeFoo establishes an effective paradigm for memory-efficient fine-tuning of large language models while maintaining competitive convergence rates. Through the integration of preconditioner-enhanced optimization with a layer-wise update strategy, PeFoo-L achieves a 2.73× reduction in memory footprint compared to Adam and outperforms existing zeroth-order methods, MeZO and HiZOO, in step-wise convergence speed. Theoretical analyses validate the convergence guarantees of both PeFoo and its layer-wise variant, PeFoo-L, under practical assumptions of smoothness and bounded curvature. The decoupling of preconditioner computation from full-model updates enables PeFoo-L to bridge the gap between memory efficiency and optimization efficacy—a critical advancement for edge-device deployment and offload the design effort of backpropagation on a hardware accelerator.

While PeFoo-L significantly reduces wall-clock training time through selective layer activation, its performance under extreme parameter sparsity or combined with advanced techniques like mixed-precision quantization remains an open question. Future work will explore integrating PeFoo with emerging sparse training frameworks and designing a specific hardware accelerator.

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

## A HYPERPARAMETERS FOR EXPERIMENTS

The hyperparameters for all PeFoo and PeFoo-L experiments on the RoBERTa-large and OPT-1.3B models are detailed in Table 5. An empirical observation from our tuning process is that the optimal learning rate for the layer-wise PeFoo-L is consistently approximately an order of magnitude larger than that for PeFoo. This finding aligns with similar phenomena reported in related papers on layer-wise like optimization, such as B-PDF Yu et al. (2024).

## B FORMULA OF UNBIASED HESSIAN MATRIX ESTIMATION

We first assume preconditioner $D$ is a symmetric positive semi-definite matrix. We denote $D^{-1/2}\nabla^2\mathcal{L}(\theta;\mathcal{B})D^{-1/2}$ as $A$, use the nature of normal distribution and apply the outer product transformation $D^{1/2}pp^\top D^{1/2}$ to the quadratic form $p^\top Ap$. Specifically, the (k,l)-th element of the reconstructed matrix $\mathbb{E}\left[p^\top D^{-1/2}\nabla^2\mathcal{L}(\theta;\mathcal{B})D^{-1/2}p \cdot D^{1/2}pp^\top D^{1/2}\right]$ is obtained through this operation:

$$
\mathbb{E}\left[p^\top D^{-1/2}\nabla^2\mathcal{L}(\theta;\mathcal{B})D^{-1/2}p \cdot D^{1/2}pp^\top D^{1/2}\right]_{k,l}
$$
$$
= \sum_{i,j,m,n} A_{i,j}D_{km}^{1/2}D_{ln}^{1/2}E\left[p_ip_jp_mp_n\right]
$$
$$
= \sum_{i,j,m,n} A_{i,j}D_{km}^{1/2}D_{ln}^{1/2}(\delta_{ij}\delta_{mn}+\delta_{im}\delta_{jn}+\delta_{in}\delta_{jm})
$$
$$
= \mathrm{tr}\left(A\right)\cdot D_{kl}+2\left[D^{\frac{1}{2}}AD^{\frac{1}{2}}\right]_{kl}, \tag{4}
$$

where $\delta_{ij}$ denotes the Kronecker delta symbol ($\delta_{ij}=1$ if $i=j$ and 0 otherwise). The final equality utilizes the symmetric property of matrix $D$ and leverages the fundamental relationship $\mathbb{E}[p_ip_jp_mp_n]=\delta_{ij}\delta_{mn}+\delta_{im}\delta_{jn}+\delta_{in}\delta_{jm}$ for standard normal variables. Expressed in matrix notation, Eq. 4 yields the closed-form relationship:

$$
\mathbb{E}\left[p^\top D^{-1/2}\nabla^2\mathcal{L}(\theta;\mathcal{B})D^{-1/2}p \cdot D^{1/2}pp^\top D^{1/2}\right] = \mathrm{tr}\left(HD^{-1}\right)D+2H.
$$

The expectation term $\mathbb{E}\left[p^\top D^{-1/2}\nabla^2\mathcal{L}(\theta;\mathcal{B})D^{-1/2}p \cdot D\right]$ provides an unbiased estimator for the trace-term product $\mathrm{tr}\left(HD^{-1}\right)D$. Therefore,

$$
\widehat{H}=\nabla^2\mathcal{L}(\theta)=\frac{1}{2}\left(\left(\mathrm{tr}\left(HD^{-1}\right)D+2H\right)-\mathrm{tr}\left(HD^{-1}\right)D\right)
$$
$$
=\frac{1}{2}\mathbb{E}\left[p^\top D^{-1/2}\nabla^2\mathcal{L}(\theta;\mathcal{B})D^{-1/2}p\cdot\left(D^{1/2}pp^\top D^{1/2}-D\right)\right].
$$

Following HiZOO Zhao et al. (2025), through Taylor's expansion, we yield the following results:

$$
\mathcal{L}\left(\theta+\lambda D^{-1/2}p;\mathcal{B}\right)=\mathcal{L}(\theta;\mathcal{B})\pm\lambda\left\langle\nabla\mathcal{L}(\theta;\mathcal{B}),D^{-1/2}p\right\rangle+\frac{\lambda^2}{2}p^\top D^{-1/2}\nabla^2\mathcal{L}(\theta;\mathcal{B})D^{-1/2}p+\mathcal{O}\left(\lambda^3\right).
$$

Then we can calculate the difference $\Delta\mathcal{L}$ by:

$$
\Delta\mathcal{L}=\mathcal{L}(\theta+\lambda D^{-1/2}p;\mathcal{B})+\mathcal{L}(\theta-\lambda D^{-1/2}p;\mathcal{B})-2\mathcal{L}(\theta;\mathcal{B})
$$
$$
=\lambda^2 p^\top D^{-1/2}\nabla^2\mathcal{L}(\theta;\mathcal{B})D^{-1/2}p+\mathcal{O}\left(\lambda^3\right),
$$

Therefore, we derive the Hessian approximation at parameter configuration $\theta$ through the following equation:

$$
\widehat{H}=\nabla^2\mathcal{L}(\theta)=\frac{1}{2}\left[\frac{\Delta\mathcal{L}}{\lambda^2}\cdot\left(D^{1/2}pp^\top D^{1/2}-D\right)\right].
$$

Table 5: The hyperparameter grids used for RoBERTa-large and OPT-1.3B experiments.

| Method | Hyperparameters | Roberta-large Values | OPT-1.3B Values |
|---|---|---|---|
| PeFoo | Batch size | 64 | 16 |
| | Learning rate | {1e-6, 2e-6} | {1e-7, 2e-7, 3e-7} |
| | $\lambda$ | 1e-3 | 1e-3 |
| | Weight Decay | 0 | 0 |
| | EMA Type | Constant 1e-5 | Constant 1e-5 |
| PeFoo | Batch size | 64 | 16 |
| (prefix) | Learning rate | {3e-3, 5e-3} | {1e-3, 3e-3, 5e-3} |
| | $\lambda$ | 1e-1 | 1e-1 |
| | Weight Decay | 0 | 0 |
| | EMA Type | Constant 1e-2 | Constant 1e-1, Constant 1e-2 |
| | # prefix tokens | 5 | 5 |
| PeFoo | Batch size | 64 | 16 |
| (LoRA) | Learning rate | {1e-4, 2e-4} | {1e-6, 5e-6, 8e-6} |
| | $\lambda$ | 1e-3 | 1e-3 |
| | Weight Decay | 0 | 0 |
| | $(r, \alpha)$ | (8, 16) | (8, 16) |
| | EMA Type | Constant 1e-4 | Constant 1e-4 |
| PeFoo-L | Batch size | 64 | 16 |
| | Learning rate | {5e-5, 7e-5, 1e-4} | {1e-6, 3e-6, 5e-6} |
| | $\lambda$ | 1e-3 | 1e-3 |
| | Weight Decay | 0 | 0 |
| | EMA Type | Constant 1e-4 | Constant 1e-4 |
| | Layer-wise Update Strategy | Select next layer every 10 steps | Select next layer every 40 steps |
| PeFoo-L | Batch size | 64 | 16 |
| (prefix) | Learning rate | {5e-2, 1e-1, 2e-1, 3e-1} | {7e-2, 1e-1, 1e-2} |
| | $\lambda$ | 1e-1 | 1e-1 |
| | Weight Decay | 0 | 0 |
| | EMA Type | Constant 1e-1, Constant 1e-2 | Constant 1e-2 |
| | Layer-wise Update Strategy | Select next layer every 10 steps | Select next layer every 40 steps |
| | # prefix tokens | 5 | 5 |
| PeFoo-L | Batch size | 64 | 16 |
| (LoRA) | Learning rate | {5e-3, 7e-3, 1e-2} | {1e-4, 3e-4, 5e-4} |
| | $\lambda$ | 1e-3 | 1e-3 |
| | Weight Decay | 0 | 0 |
| | $(r, \alpha)$ | (8, 16) | (8, 16) |
| | EMA Type | Constant 1e-4 | Constant 1e-4 |
| | Layer-wise Update Strategy | Select next layer every 10 steps | Select next layer every 40 steps |
| Adam | Batch size | {2, 4, 8} | {2, 4, 8} |
| | Learning rate | {1e-5, 3e-5, 5e-5} | {1e-6, 3e-6, 5e-6} |

## C PeFoo Convergence Analysis

We present a brief proof of the convergence of PeFoo. Our convergence analysis relies on the following **assumptions**:

1. The objective function $\mathcal{L}(\theta)$ is $L$-smooth, i.e. $\mathcal{L}(\theta_{t+1}) \leqslant \mathcal{L}(\theta_t) - \langle \nabla \mathcal{L}(\theta_t), \theta_{t+1} - \theta_t \rangle + \frac{L}{2} \|\theta_{t+1} - \theta_t\|^2$;

2. The stochastic gradient $\nabla \mathcal{L}(\theta; \mathcal{B})$ has $\sigma^2$ variance, i.e. $\mathbb{E}\left[\|\nabla \mathcal{L}(\theta; \mathcal{B}) - \nabla \mathcal{L}(\theta)\|^2\right] \leqslant \sigma^2$;

3. $D = \text{diag}(d_1, \ldots, d_n)$, with $0 < \beta_\ell \leqslant d_i \leqslant \beta_u$.

*Proof.* According to Taylor's expansion, we have:

$$\widehat{\nabla}\mathcal{L}(\theta_t) = \frac{\mathcal{L}(\theta_t + \lambda D_t^{-1/2}p) - \mathcal{L}(\theta_t - \lambda D_t^{-1/2}p)}{2\lambda} D_t^{-1/2}p,$$

$$= D_t^{-1/2}pp^\top D_t^{-1/2}\nabla\mathcal{L} + \mathcal{O}(\lambda^2).$$

Next, we compute the expectation of the squared norm of $\widehat{\nabla}\mathcal{L}(\theta_t)$:

$$\mathbb{E}\left[\|\widehat{\nabla}\mathcal{L}(\theta_t)\|^2\right] = \mathbb{E}\left[\nabla\mathcal{L}^\top D^{-1/2}pp^\top D^{-1}pp^\top D^{-1/2}\nabla\mathcal{L}\right]$$

$$=\mathbb{E}\left[\sum_{k=1}^n d_k^{-2}p_k^4(\nabla\mathcal{L}_k)^2 + \sum_{k\neq l}d_k^{-1}d_l^{-1}p_k^2p_l^2\nabla\mathcal{L}_k\nabla\mathcal{L}_l\right]$$

$$=3\sum_{k=1}^n d_k^{-2}(\nabla\mathcal{L}_k)^2 + \sum_{k\neq l}d_k^{-1}d_l^{-1}\nabla\mathcal{L}_k\nabla\mathcal{L}_l$$

$$=2\sum_{k=1}^n d_k^{-2}(\nabla\mathcal{L}_k)^2 + \left(\sum_{k=1}^n d_k^{-1}\nabla\mathcal{L}_k\right)^2$$

$$\leqslant 2\beta_l^{-1}\|\nabla\mathcal{L}\|_{D_t^{-1}}^2 + \mathrm{tr}(D_t^{-1})\|\nabla\mathcal{L}\|_{D_t^{-1}}^2 = C_1\|\nabla\mathcal{L}\|_{D_t^{-1}}^2, \qquad (5)$$

where $C_1 = 2\beta_l^{-1} + \mathrm{tr}(D_t^{-1}) \leqslant (n+2)\beta_l^{-1}$. Using the update rule for $\theta_t$ and the assumptions above, we obtain the following bound on the expected change in the objective function:

$$\mathbb{E}[\mathcal{L}(\theta_{t+1};\mathcal{B})] - \mathbb{E}[\mathcal{L}(\theta_t;\mathcal{B})]$$

$$\leqslant -\eta_t\mathbb{E}\left[\langle\nabla\mathcal{L}(\theta_t;\mathcal{B}), \widehat{\nabla}\mathcal{L}(\theta_t;\mathcal{B})\rangle\right] + \frac{L\eta_t^2}{2}\mathbb{E}\left[\|\widehat{\nabla}\mathcal{L}(\theta_t;\mathcal{B})\|^2\right]$$

$$\leqslant -\eta_t\|\nabla\mathcal{L}\|_{D_t^{-1}}^2 + \mathcal{O}(\lambda^2) + \frac{L\eta_t^2}{2}\left(C_1\|\nabla\mathcal{L}\|_{D_t^{-1}}^2 + \sigma^2\right) \qquad (6)$$

$$\leqslant -\frac{\eta_t}{2}\|\nabla\mathcal{L}\|_{D_t^{-1}}^2 + \frac{L\eta_t^2\sigma^2}{2} + \mathcal{O}(\lambda^2),$$

where the last inequality is because we choose $\eta_t = \eta \leqslant \frac{1}{LC_1}$. Summing over $t = 0, 1, ..., T-1$, we obtain:

$$\sum_{t=0}^{T-1}\frac{\eta_t}{2}\|\nabla\mathcal{L}\|_{D_t^{-1}}^2 \leqslant \mathbb{E}[\mathcal{L}(\theta_0;\mathcal{B}) - \mathcal{L}(\theta_T;\mathcal{B})] + \frac{L\eta_t^2\sigma^2 T}{2} + \mathcal{O}(\lambda^2)$$

$$\leqslant \mathbb{E}[\mathcal{L}(\theta_0;\mathcal{B}) - \mathcal{L}^*] + \frac{L\eta_t^2\sigma^2 T}{2} + \mathcal{O}(\lambda^2).$$

Rearranging and summing over the $T$ iterations, we have:

$$\mathbb{E}\left[\|\nabla\mathcal{L}(\theta;\mathcal{B})\|^2\right] \leqslant \mathbb{E}\left[\frac{1}{T}\sum_{t=1}^T\|\nabla\mathcal{L}(\theta_t;\mathcal{B})\|^2\right]$$

$$\leqslant \frac{\beta_u}{T}\sum_{t=1}^T\|\nabla\mathcal{L}(\theta_t;\mathcal{B})\|_{D_t^{-1}}^2 \leqslant \frac{2\beta_u(\mathcal{L}(\theta_0)-\mathcal{L}^*)}{\eta T} + L\beta_u\eta\sigma^2 + \mathcal{O}(\lambda^2).$$

Let $\eta \to 0$ as $T \to \infty$ (e.g., $\eta = \mathcal{O}(1/\sqrt{T})$), we conclude that:

$$\lim_{T\to\infty}\mathbb{E}\left[\|\nabla\mathcal{L}(\theta;\mathcal{B})\|^2\right] = 0. \qquad \square$$

## D PEFOO-L ALGORITHM AND CONVERGENCE ANALYSIS

Details of PeFoo-L can be seen in Algorithm 2. Considering that layer-wise weight updates can lead to cyclic behavior and hinder convergence, we provide a proof for PeFoo-L.

---

**Algorithm 2** Training Pipeline of PeFoo-L.

---

**Require:** parameters $\theta \in \mathbb{R}^d$, loss function $\mathcal{L}$, perturbation scale $\lambda$, learning rate $\eta$, smooth scale $\alpha$
1: **for** $t = 1, \ldots, T$ **do**
2:     Sample a random seed $s$
3:     Select optimizing current layer or next layer
4:     **if** not select current $(j-1)$-th layer **then**
5:         Remove $D_{t-1}^{(j-1)}$ and $D_{t-1}^{(j)} \leftarrow I$                    $\triangleright$ Preconditioner Initialization
6:     **end if**
7:     $\ell \leftarrow \mathcal{L}(\theta; \mathcal{B})$
8:     $\theta \leftarrow \text{PerturbParameters}(\theta, \lambda, (D_{t-1}^{(j)})^{-1/2}, s)$
9:     $\ell_\lambda \leftarrow \mathcal{L}(\theta; \mathcal{B})$
10:    $\theta \leftarrow \text{PerturbParameters}(\theta, -2\lambda, (D_{t-1}^{(j)})^{-1/2}, s)$
11:    $\ell_{-\lambda} \leftarrow \mathcal{L}(\theta; \mathcal{B})$
12:    $\theta \leftarrow \text{PerturbParameters}(\theta, \lambda, (D_{t-1}^{(j)})^{-1/2}, s)$          $\triangleright$ Reset Parameters Before Descent
13:    projected_grad $\leftarrow \frac{1}{2\lambda}(\ell_\lambda - \ell_{-\lambda})(D_{t-1}^{(j)})^{-1/2}$
14:    Reset random number generator with seed $s$
15:    **for** $\theta_i \in$ selected layer **do**
16:        Sample $p_i \sim \mathcal{N}_s(0, I)$
17:        $\theta_i \leftarrow \theta_i - \eta_t \times \text{projected\_grad}_i \times p_i$            $\triangleright$ Update Weights
18:    **end for**
19:    $\hat{H}_t \leftarrow \frac{1}{2\lambda^2}(\ell_\lambda + \ell_{-\lambda} - 2\ell)\left((D_{t-1}^{(j)})^{1/2} p_i p_i^\top (D_{t-1}^{(j)})^{1/2} - D_{t-1}^{(j)}\right)$
20:    $H_t \leftarrow (1 - \alpha_t)H_{t-1} + \alpha_t \hat{H}_t$
21:    $(D_t^{(j)}) \leftarrow abs(H_t)$               $\triangleright$ Update Preconditioner After Weights Update
22: **end for**

23: **function** PerturbParameter$(\theta, \lambda, (D_t^{(j)})^{-1/2}, s)$
24:    Reset random number generator with seed $s$
25:    **for** $\theta_i \in$ selected layer **do**
26:        Sample $p_i \sim \mathcal{N}_s(0, I)$
27:        $\theta_i \leftarrow \theta_i + \lambda(D_t^{(j)})^{-1/2} p_i$            $\triangleright$ Perturb Parameters With Preconditioner
28:    **end for**
29:    **return** $\theta$
30: **end function**

---

In the $t$-th iteration, the update for the $i$-th layer is given by: $\theta_{t+1}^{(i)} = \theta_t^{(i)} - \eta_t^{(i)} \widehat{\nabla} \mathcal{L}(\theta_t^{(i)}; \mathcal{B}_t^{(i)})$, where $\widehat{\nabla} \mathcal{L}(\theta_t^{(i)}; \mathcal{B}_t^{(i)})$ is a gradient estimate of $\theta_t^{(i)}$ based on parameter $\theta_{t+1}^{<i} = (\theta_{t+1}^{(1)}, \ldots, \theta_{t+1}^{(i-1)}, \theta_t^{(i)}, \theta_t^{(i+1)}, \ldots, \theta_t^{(N)})$.

We add the following **assumptions**:

1. Layer-wise smoothness. There is a constant $M$ that $\|\nabla_{\theta^{(i)}} \mathcal{L}(\theta_t) - \nabla_{\theta^{(i)}} \mathcal{L}(\theta_{t+1}^{<i-1})\| \leqslant M\|\theta_t - \theta_{t+1}^{<i-1}\|$.

2. For every layer $i$, the variance of gradient have: $\mathbb{E}\left[\|\widehat{\nabla} \mathcal{L}(\theta^{(i)}) - D_t^{-1} \nabla_{\theta^{(i)}} \mathcal{L}(\theta)\|^2\right] \leqslant \sigma_i^2$.

*Proof.* Due to interlayer coupling, updating the $i$-th layer affects the input distribution of subsequent layers. Define global parameter changes: $\theta_{t+1} = \theta_t - \sum_{i=1}^N \eta_t^{(i)} \widehat{\nabla} \mathcal{L}(\theta_t^{(i)}; \mathcal{B}_t^{(i)}) \cdot e_i$, where $e_i$ is the unit vector of the $i$-th layer. Use the overall L-smoothness as well as Eq. 6:

$$\mathbb{E}[\mathcal{L}(\theta_{t+1})] - \mathbb{E}[\mathcal{L}(\theta_t)] \leqslant -\sum_{i=1}^N \frac{\eta_t^{(i)}}{2} \|\nabla_{\theta^{(i)}} \mathcal{L}(\theta_t)\|_{D_t^{-1}}^2 + \sum_{i=1}^N \frac{L(\eta_t^{(i)})^2 \sigma_i^2}{2} + \Delta_{\text{coupling}}, \quad (7)$$

where $C_1^{(i)} = 2\beta_l^{-1} + \text{tr}(D_t^{-1})$, $\eta_t^{(i)} \leqslant \frac{1}{LC_1^{(i)}}$, and $\Delta_{\text{coupling}}$ accounts for the term introduced by the interlayer coupling:

$$\Delta_{\text{coupling}} = \mathbb{E}\left[\sum_{i=1}^{N} \eta_t^{(i)} \left\langle \nabla_{\theta^{(i)}} \mathcal{L}(\theta_t), D_t^{-1} \nabla_{\theta^{(i)}} \mathcal{L}(\theta_t) - \widehat{\nabla}\mathcal{L}(\theta_t^{(i)}) \right\rangle\right].$$

According to assumption 1, we have:

$$\|\nabla_{\theta^{(i)}} \mathcal{L}(\theta_t) - \nabla_{\theta^{(i)}} \mathcal{L}(\theta_{t+1}^{<i-1})\| \leqslant M\|\theta_t - \theta_{t+1}^{<i-1}\| = M \sum_{j=1}^{i-1} \eta_t^{(j)} \|\widehat{\nabla}\mathcal{L}(\theta_t^{(j)})\|.$$

Next, we analyze each term of $\Delta_{\text{coupling}}$:

$$\mathbb{E}\left[\left\langle \nabla_{\theta^{(i)}} \mathcal{L}(\theta_t), D_t^{-1} \nabla_{\theta^{(i)}} \mathcal{L}(\theta_t) - \widehat{\nabla}\mathcal{L}(\theta_t^{(i)}) \right\rangle\right] =$$

$$\mathbb{E}\left[\left\langle \nabla_{\theta^{(i)}} \mathcal{L}(\theta_t), D_t^{-1} \left(\nabla_{\theta^{(i)}} \mathcal{L}(\theta_t) - \nabla_{\theta^{(i)}} \mathcal{L}(\theta_{t+1}^{<i-1})\right)\right\rangle\right]$$

$$\leqslant \|\nabla_{\theta^{(i)}} \mathcal{L}(\theta_t)\|_{D_t^{-1}} \cdot \mathbb{E}\left[\|\nabla_{\theta^{(i)}} \mathcal{L}(\theta_t) - \nabla_{\theta^{(i)}} \mathcal{L}(\theta_{t+1}^{<i-1})\|_{D_t^{-1}}\right]$$

$$\leqslant \|\nabla_{\theta^{(i)}} \mathcal{L}(\theta_t)\|_{D_t^{-1}} \cdot \beta_\ell^{-1/2} M \sum_{j=1}^{i-1} \eta_t^{(j)} \mathbb{E}\left[\|\widehat{\nabla}\mathcal{L}(\theta_t^{(j)})\|\right]$$

$$\leqslant \|\nabla_{\theta^{(i)}} \mathcal{L}(\theta_t)\|_{D_t^{-1}} \cdot \beta_\ell^{-1/2} M \sum_{j=1}^{i-1} \eta_t^{(j)} \sqrt{C_1^{(j)} \|\nabla_{\theta^{(j)}} \mathcal{L}(\theta_t)\|_{D_t^{-1}}^2 + \sigma_j^2}$$

$$\leqslant \frac{1}{4} \|\nabla_{\theta^{(i)}} \mathcal{L}(\theta_t)\|_{D_t^{-1}}^2 + M^2 \beta_\ell^{-1}(i-1) \sum_{j=1}^{i-1} (\eta_t^{(j)})^2 \left(C_1^{(j)} \|\nabla_{\theta^{(j)}} \mathcal{L}(\theta_t)\|_{D_t^{-1}}^2 + \sigma_j^2\right),$$

where the first inequality is because of Cauchy's Inequality, the third inequality uses Eq. 5, and the last inequality is because of Young's Inequality. We select $\eta$ such that:

$$\eta_t^{(i)} = \eta = \frac{1}{\sqrt{T} \max\{Lmax_i C_1^{(i)}, \sqrt{8M^2 \beta_\ell^{-1} N \max_j C_1^{(j)}}\}},$$

so that the second term can be absorbed into the main descent term:

$$M^2 \beta_\ell^{-1} \sum_{i=1}^{N} \eta(i-1) \sum_{j=1}^{i-1} \eta^2 \left(C_1^{(j)} \|\nabla_{\theta^{(j)}} \mathcal{L}(\theta_t)\|_{D_t^{-1}}^2 + \sigma_j^2\right) \leqslant \frac{1}{8} \sum_{i=1}^{N} \eta \|\nabla_{\theta^{(i)}} \mathcal{L}(\theta_t)\|_{D_t^{-1}}^2 + \mathcal{O}\left(\frac{\eta \sigma_{\text{total}}^2}{T}\right).$$

Substituting $\Delta_{\text{coupling}}$ into the Eq. 7, we obtain:

$$\sum_{t=0}^{T-1} \sum_{i=1}^{N} \frac{\eta}{8} \|\nabla_{\theta^{(i)}} \mathcal{L}(\theta_t)\|_{D_t^{-1}}^2 \leqslant \mathcal{L}(\theta_0) - \mathcal{L}^* + \frac{L\eta^2 \sigma_{\text{total}}^2 T}{2}.$$

Since $\eta = \mathcal{O}(1/\sqrt{T})$ and $d_i \leqslant \beta_u$, we conclude:

$$\frac{1}{T} \sum_{t=0}^{T-1} \sum_{i=1}^{N} \|\nabla_{\theta^{(i)}} \mathcal{L}(\theta_t)\| \leqslant \mathcal{O}\left(\frac{1}{\sqrt{T}}\right). \qquad \square$$

# E THE USE OF LARGE LANGUAGE MODELS

We use LLM only to polish our writing.

