# OpenReview forum: "PeFoo-L: A General Framework for Preconditioned Enhanced Forward-Only Optimizer in LLM Fine-tuning on the Edge"
_ICLR.cc/2026/Conference — ICLR 2026 Conference Withdrawn Submission_

### Official Review · Reviewer_pp3x · 2025-10-18

**Soundness:** 2
**Presentation:** 3
**Contribution:** 2
**Rating:** 4
**Confidence:** 4

**Summary:**

The paper proposes PeFoo and PeFoo-L, a preconditioned forward-only optimization framework for fine-tuning large language models under constrained memory settings. The method introduces a diagonal preconditioner within a zeroth-order optimization scheme to better adapt to curvature information, and further employs a layer-wise update strategy to reduce memory and bandwidth overhead.

**Strengths:**

- The proposed PeFoo-L variant is practical, as it significantly reduces memory and storage costs while maintaining competitive performance.

- Theoretical analysis is provided to support convergence and stability, adding rigor to the proposed approach.

**Weaknesses:**

- The degree of novelty seems limited. The method is essentially an extension of HiZOO with refinements and engineering choices.

- Experiments are primarily conducted on relatively older models (e.g., RoBERTa, OPT-1.3B). It would strengthen the paper to evaluate on more recent LLMs to demonstrate broader applicability.

- HiZOO also introduced HiZOO-L version. How does your method differ in principle or in practice, and what concrete advantages does it bring over HiZOO-L? A direct comparison and deeper discussion would clarify the contribution.

- It would be very useful to include more loss curves for different models and datasets to demonstrate training dynamics more clearly.

**Questions:**

Seen in weakness

---

### Official Review · Reviewer_UY7c · 2025-10-31

**Soundness:** 2
**Presentation:** 3
**Contribution:** 2
**Rating:** 4
**Confidence:** 3

**Summary:**

The paper proposes PeFoo a preconditioned forward‑only ZO optimizer for on‑device fine‑tuning of LLMs, and PeFoo‑L, a memory‑ and bandwidth‑aware layer‑wise variant. In PeFoo, the standard MeZO/SPSA estimator is reparameterized with a diagonal preconditioner D, so perturbations become anisotropic $\lambda D^{-1/2}p$. A key technical element is a moment‑based, trace‑corrected Hessian estimator that uses the same three forward passes per step to form
$
\hat H ;=; \tfrac12,\frac{\Delta L}{\lambda^2},\big(D^{1/2}pp^\top D^{1/2}-D\big),\qquad
\Delta L=L(\theta+\lambda D^{-1/2}p)+L(\theta-\lambda D^{-1/2}p)-2L(\theta),
$
and then builds a diagonal preconditioner via $D=\mathrm{clip}(|\mathrm{EMA}(\hat H)|, D_{\min}, D_{\max})$. The paper argues this fixes two issues claimed for HiZOO (biased magnitude accumulation and an update inconsistency), and gives a convergence bound under smoothness and bounded‑$D$. To reduce memory traffic and state, PeFoo‑L maintains and applies the preconditioner for only one layer per step. Experiments on OPT‑1.3B and RoBERTa‑large report accuracy with near‑MeZO memory.

**Strengths:**

1. It introduces a trace‑corrected Hessian estimator tailored to forward‑only training and a layer‑wise preconditioner that targets weight‑update bandwidth.
2. Theoretical assumptions and proofs are standard and mostly sound; using $D_{t-1}$ consistently avoids the look‑ahead inconsistency flagged for HiZOO.
3. Algorithms and figures are easy to follow.

**Weaknesses:**

1. App. B mixes central‑difference with an $O(\lambda^3)$ remainder. Central schemes eliminate odd terms, so the leading error is $O(\lambda^4)$. “Unbiased” should be qualified as “for the smoothed loss, up to small $\lambda$”.
2. The proof hides ZO variance in a generic $\sigma^2$ and introduces constants like $C_1=2\beta_\ell^{-1}+\mathrm{tr}(D^{-1})$, implying explicit $d$‑dependence in stepsize conditions. This needs discussion or a lemma bounding $\mathbb{E}|\nabla L_p|^2$ under the chosen $D$.
3. Comparisons vs. Adam are not compute‑ or wall‑clock‑matched (e.g., 20k ZO steps vs. 5 epochs on 1k examples). It makes PeFoo vs. FT conclusions hard to interpret. Similar step‑budget mismatches appear in Table 4.
4. The paper clamps HiZOO’s Hessian at $10^4$ to prevent overflow, which may change its behavior; FP32/AMP results would isolate numerical causes from algorithmic differences.
5. Alg. 2 reinitializes $D$ when switching layers, discarding history; no ablation quantifies whether caching last‑seen preconditioners helps or hurts memory/speed.
6. The claim that PeFoo “preserves negative curvature information” is true within EMA, but $|\cdot|$ is applied at the end; please nuance the claim. Also clarify RNG reuse between perturb and update in Alg. 1.
7. Other related works that might be worth comparing regarding differences including - HELENE (EMNLP 2025), ReLIZO (NeurIPS 2024), LOZO ICLR 2025.

**Questions:**

1. For the central two‑sided difference in Eq. (2), shouldn’t $\Delta L=\lambda^2 p^\top D^{-1/2}HD^{-1/2}p + O(\lambda^4)$ (not $O(\lambda^3))$? If so, your bias term in Sec. 3.2 becomes $O(\lambda^4)$.
2. Can you bound $\mathbb{E}|\nabla L_p|^2$ for the preconditioned estimator in terms of $D$, $d$, and $|\nabla L|$? This would clarify step‑size choices and the role of $\mathrm{tr}(D^{-1})$.
3. Please add accuracy vs. forward‑evaluations (plots for Adam vs. MeZO/HiZOO/PeFoo/PeFoo‑L on SST‑2 and SQuAD to address the step/epoch mismatches in Tables 3–4.
4. Do FP32/AMP runs of HiZOO (even at higher memory) show similar gaps? How sensitive is PeFoo to (D_{\max}) in FP16?
5. What happens if you cache the preconditioner for the $k$ most‑recent layers instead of reinitializing to $I$? Is there a sweet spot between memory and speed?
6.  Which Nsight metrics underpin Table 2 and Fig. 4? Were GPU clocks fixed? Please include batch size and whether numbers include cache line write‑backs.
7. A small sweep of perturbation scale $\lambda$ would inform the $O(\lambda^2)$/variance trade‑off assumed in Sec. 3.2.
8. Did you try block‑diagonal $D$ (e.g., per‑matrix blocks)? Even one experiment could show whether extra structure is worth the added traffic.

---

### Official Review · Reviewer_8XKo · 2025-11-01

**Soundness:** 2
**Presentation:** 2
**Contribution:** 2
**Rating:** 4
**Confidence:** 4

**Summary:**

This paper proposes an algorithm PeFoo for the preconditioned zeroth-order (ZO) optimization in LLM training. Previous work HiZOO has adopted the estimated Hessian information into the ZO gradient to accelerate the convergence. However, some fundamental issues lead HiZOO to converge in suboptimal paths. In contrast, PeFoo designs an unbiased estimation of the Hessian matrix to perform the EMA update coupled with clipping. Additionally, it utilizes the Hessian matrix in the last iteration instead of the fresh version in the current iteration to calculate the projected gradient.

This work then provides the convergence guarantee for PeFoo. In the experiments of fine-tuning two LLMs, an extension version PeFoo-L that adopts the layerwise update rule is proposed to further reduce the memory cost. PeFoo outperforms baselines in most downstream tasks, and PeFoo-L significantly reduces the GPU usage. Further studies show that PeFoo-L indeed reduces the memory cost of weight update, and the techniques in preconditioner clamping are effective.

**Strengths:**

The proposed approach has two main innovations: one is an unbiased Hessian estimator with EMA and clipping, and another is the utilization of the past preconditioner. The experimental results reveal that PeFoo could enhance the performances of MeZO and HiZOO. The ablation studies are also detailed. Thus, these evidences could support the claims of the paper. The presentation of this work is also clear and easy to understand.

**Weaknesses:**

Some statements lack further explanation or support from references. For instance, it is said in "Introduction" that “our analysis reveals that HiZOO’s Hessian estimator is inherently biased due to the premature application of an absolute value function”, but I do not find the related analysis in the following chapters.

The idea of designing the algorithm is also not clear. It is said that “HiZOO’s estimation tends to be larger than PeFoo’s unbiased estimation and does not account for the possibility of negative elements in the Hessian matrix, leading to saddle point attraction artifacts”. Firstly, how to understand that “estimation tends to be larger”? Secondly, this assertion is not analyzed or explained. Then for the “weight update step”, this work also does not make detailed explanation for “This inconsistency is the reason for the large variance in HiZOO’s Hessian matrix estimation”. Thus, the motivation of proposing such an algorithm and the reason it performs well are not clear. In addition, the colors of the curves in Figures 2 and 3 are too similar to distinguish.

**Questions:**

In Figure 1, it shows that MeZO performs two perturbations in one iteration. Is this right? I think MeZO seem to also perform three perturbations.

---

### Official Review · Reviewer_iag3 · 2025-11-01

**Soundness:** 2
**Presentation:** 3
**Contribution:** 2
**Rating:** 4
**Confidence:** 3

**Summary:**

The paper introduces PeFoo, a preconditioner‑enhanced forward‑only (ZO/SPSA) optimizer for fine‑tuning LLMs under tight memory budgets, and PeFoo‑L, a layer‑wise variant that keeps only one layer’s preconditioner active per step to cut storage and DRAM traffic. Methodologically, PeFoo injects a diagonal preconditioner D into both perturbation and projected‑gradient formulas， and claims a corrected, “unbiased” Hessian estimator plus a weight‑update rule consistent with the estimator, addressing instability of HiZOO.

**Strengths:**

1. Preconditioner D shapes both perturbations and projected gradients, aligning exploration with local curvature.

2. The paper corrects HiZOO’s estimator/weight‑update mismatch and preserves Hessian sign before taking absolute value post‑EMA, which the ablation shows improves stability and accuracy.

3. Meaningful empirical gains under FP16, and shows concrete peak‑memory savings vs Adam/HiZOO

**Weaknesses:**

1. The derivation hinges on second‑order central differences with an O(\lambda^3) remainder， but Sec. 3.1 repeatedly calls the estimator “unbiased” and builds claims on it. In practice the estimate is for L(\theta;B), not the population objective, and the bias/variance from minibatching is never analyzed.

2. The paper advertises preserving negative curvature (p. 2), but the preconditioner actually used in updates is PSD by construction. Signs exist only inside EMA and are discarded before use; no theory is offered for why “abs after EMA” should beat “abs before EMA” beyond a single ablation.

3. H^t depends on 𝐷_(t-1}, Alg. 1 line 16); Dt is then built from H^t . This feedback could amplify noise and cause oscillations. The paper relies on hand‑picked clamps 𝐷min=0.1 and 𝐷max=10^4,  and a single‑task curve, with no stability analysis.

4. the paper claims O(d) memory by using a diagonal D, but according to algorithm 1, implementation is O(d^2).

5. Alg. 2 resets the preconditioner to 𝐼, when switching layers (line 5), repeatedly erasing accumulated geometry; the “last‑to‑first” schedule and “switch every 40 steps” are empirical only, with no principled rationale or ablations.

6. PeFoo‑L is essentially preconditioned block‑coordinate forward‑only. The paper does not compare against peer block/layer methods under the same scheduling and budget, making it unclear whether gains stem from the preconditioner itself or from the scheduling

**Questions:**

NA

---

### Note · Authors · 2025-11-17

**Comment:**

The authors thank the reviewers for their valuable suggestions.

**Withdrawal Confirmation:**

I have read and agree with the venue's withdrawal policy on behalf of myself and my co-authors.